# Development of the individualised Comparative Effectiveness of Models Optimizing Patient Safety and Resident Education (iCOMPARE) trial: a protocol summary of a national cluster-randomised trial of resident duty hour policies in internal medicine

Judy A Shea,[1] Jeffrey H Silber,[2] Sanjay V Desai,[3] David F Dinges,[4] Lisa M Bellini,[1] James Tonascia,[5] Alice L Sternberg,[6] Dylan S Small,[7] David M Shade,[6] Joel Thorp Katz,[8] Mathias Basner,[4] Krisda H Chaiyachati,[1,9] Orit Even-Shoshan,[2] David Westfall Bates,[8] Kevin G Volpp,[9,10] David A Asch,[1,9] the iCOMPARE Research Group

For numbered affiliations see end of article.

**Correspondence to**
Dr Judy A Shea;
sheaja@pennmedicine.upenn.edu

## ABSTRACT

**Introduction** Medical trainees' duty hours have received attention globally; restrictions in Europe, New Zealand and some Canadian provinces are much lower than the 80 hours per week enforced in USA. In USA, resident duty hours have been implemented without evidence simultaneously reflecting competing concerns about patient safety and physician education. The objective is to prospectively evaluate the implications of alternative resident duty hour rules for patient safety, trainee education and intern sleep and alertness.

**Methods and analysis** 63 US internal medicine training programmes were randomly assigned 1:1 to the 2011 Accreditation Council for Graduate Medical Education resident duty hour rules or to rules more flexible in intern shift length and number of hours off between shifts for academic year 2015–2016. The primary outcome is calculated for each programme as the difference in 30-day mortality rate among Medicare beneficiaries with any of several prespecified principal diagnoses in the intervention year minus 30-day mortality in the preintervention year among Medicare beneficiaries with any of several prespecified principal diagnoses. Additional safety outcomes include readmission rates, prolonged length of stay and costs. Measures derived from trainees' and faculty responses to surveys and from time-motion studies of interns compare the educational experiences of residents. Measures derived from wrist actigraphy, subjective ratings and psychomotor vigilance testing compare the sleep and alertness of interns. Differences between duty hour groups in outcomes will be assessed by intention-to-treat analyses.

**Ethics and dissemination** The University of Pennsylvania Institutional Review Board (IRB) approved the protocol and served as the IRB of record for 40 programmes that agreed to sign an Institutional Affiliation Agreement. Twenty-three programmes opted for a local review process.
**Trial registration number** NCT02274818; Pre-results.

## Strengths and limitations of this study

► iCOMPARE (individualized Comparative Effectiveness of Models Optimizing Patient Safety and Resident Education) is the largest randomised trial examining the impact of duty hours in internal medicine (IM) training programmes.

► The aims are broad and include patient safety, trainees' sleep and alertness and a host of education outcomes.

► The protocol draws on a number of data collection methods including real time direct observation of trainee activities, existing and study administered surveys, robust recording of sleep periods over a 14-day period and patient medical record data.

► The lag between the observation period and data release for patient medical record data is substantial.

► iCOMPARE promises narrow but substantive evidence to inform resident duty hour standards in IM and signals new interest in methodologically stronger research in medical education.

## INTRODUCTION

The long hours worked by resident physicians received some academic attention in the 1970s in USA.[1] It was not until 1984 when those long hours became publicly linked with concerns about patient safety.[2] Safety concerns

resonated with a public who found it self-evident that the often 30 hours shifts of resident physicians would lead to fatigue and that fatigue would lead to errors that would harm patients. Notably, the possible link between duty hours and patient safety was not just a US concern. Many countries began to limit hours in the 1990s. For example, New Zealand has had a limit of 72 hours a week since 1985.[3] The European Working Time Directive has included doctors in training in their limits of 48 hours per week since 2000 (but Denmark has a normal work week of 37 hours). Australia and many Canadian provinces set limits on some shift lengths given concerns for patient and trainee safety.[4] USA was a little slower to come to the table. Although regulation of duty hours across all specialties occurred first in New York State as a reaction to the Libby Zion case,[2] it was not until 2003 when the Accreditation Council for Graduate Medical Education (ACGME)—the organisation that oversees resident education in USA—limited residents to 80 hours of work per week averaged over 4 weeks and limited the length of individual shifts to 24 hours, with an additional 4 hours to safely transfer care to the next resident. Partly on the basis of an Institute of Medicine report and a trial from Brigham and Women's Hospital,[5 6] those regulations were further tightened for first year residents (interns) in 2011, limiting their maximum shift length to 16 hours. This change prompted a charged debate.[7] Proponents argued that the restrictions did not go far enough. Others argued that the regulations were overly restrictive and inflexible and harboured increased risk to patients by increasing patient handoffs.[8 9]

Meanwhile, large observational studies, using data from Medicare and the Veterans Administration across millions of hospitalisations, found essentially no difference in important patient outcomes following implementation of either the 2003[10–18] or 2011 duty hour standards.[19] Several studies associated the 2011 standards with less direct patient contact, increased perception of medical errors, increased transitions of care, decreased educational opportunities and only modestly increased quantities of sleep.[20–22] Others found no changes in trainees' educational test scores.[23] Programme directors and trainees expressed concern that the rules reduced training quality and increased rather than decreased medical errors.[24–26] The increasing recognition of the importance of supervision—with separate mandates implemented by the ACGME over the same period—added further uncertainty to the debate. In the end, a well-meaning effort to manage resident fatigue was perceived by many to promote burnout, increase handoffs, decrease educational opportunities and delay the professional maturation required to produce competent, independent physicians. Currently, the evidence available to resolve these controversies is limited to a patchwork of laboratory and in vivo studies of sleep deprivation, large scale epidemiological observations of patient outcomes, surveys of resident and educator opinions and often single-centre trials of unique residency duty hour designs that focused on workload or sleep but not patient outcomes.

In this context, we created the iCOMPARE (individualized Comparative Effectiveness of Models Optimizing Patient Safety and Resident Education) trial, a cluster-randomised trial carried out by internal medicine (IM) residency programmes in USA during the 2015–2016 academic year. Participating residency programmes were randomised to one of two groups: (1) maintain standard (STD) duty hour rules or (2) permit a more flexible (FLEX) set of duty hour rules, noted principally for removing the 16 hours shift length restriction for interns and allowing them to work up to 24 hours with an additional 4 hours for care transitions. In contrast to prior work, iCOMPARE was designed to simultaneously assess the impact of duty hour rules on patient safety, resident education and intern sleep and alertness.

While the iCOMPARE trial was being planned, leaders in surgical education developed a roughly parallel trial, the Flexibility in Duty Hours Requirements for Surgical Trainees (FIRST) trial, which was fielded in the 2014–2015 academic year. The design of the FIRST trial and its initial results have been published.[27 28] Here, we describe the design of the iCOMPARE trial.

### Funding and organisation

The iCOMPARE trial is funded primarily by the National Heart, Lung, and Blood Institute (NHLBI), with additional funding from the ACGME. The NHLBI appointed an independent Data and Safety Monitoring Board (DSMB) to advise the Institute regarding the trial's progress, monitor data quality and safeguard the interests of study participants.

### Study aims and hypotheses

Study hypotheses are presented in box 1. The primary hypothesis for the trial is that 30-day any-site patient mortality under FLEX will not exceed (will not be inferior to) 30-day patient mortality under STD, measured as the difference in difference across STD and FLEX programmes between a programme's 30-day patient mortality rate in the trial year and that rate in the pretrial year (ie, 30-day patient mortality rate at a programme during the trial year minus 30-day patient mortality rate at the programme in the pretrial year). Secondary outcomes related to patient safety include 7-day and 30-day hospital readmission rates, complication rates, the probability of a prolonged length of stay, total resource utilisation and Medicare payments.

iCOMPARE's education hypotheses are that trainees in the FLEX arm will spend more time in direct patient care and education and have greater satisfaction with educational experiences compared with their STD arm peers; that standardised test scores for interns in FLEX will not be lower than such scores for interns in the STD arm; that faculty in the FLEX arm will report greater satisfaction with their clinical teaching experiences and greater perceptions of safety, teamwork and supervision than faculty in the STD arm.

## Box 1  Study aims and hypotheses

### Primary hypothesis
**Specific aim 1:** Examine patient safety and costs of care under STD and FLEX duty hour schedules.

*H1a: (Primary hypothesis) 30-day patient mortality under FLEX will not exceed (will not be inferior to) mortality under STD.*

### Secondary hypotheses
*H1b: 7-day and 30-day hospital readmission rates under FLEX will not exceed (will not be inferior to) the rates under STD.*

*H1c: Complication rates, defined by selected AHRQ Patient Safety Indicators, under Flex will not exceed (will not be inferior to) complication rates under STD.*

*H1d: The rate of prolonged length of stay under FLEX will not exceed (will not be inferior to) the rate of prolonged length of stay under STD.*

*H1e: Total costs and Medicare payments, under FLEX will not exceed (will not be inferior to) overall costs under STD.*

**Specific aim 2:** Examine the quality of education under STD and FLEX duty hour schedules.

### Secondary hypotheses
*H2a: Interns in FLEX will spend greater time in direct patient care and education compared with interns in STD.*

*H2b: Trainees in FLEX will report greater satisfaction with their educational experience (greater ownership, greater continuity and lower burnout) than trainees in STD.*

*H2c: Faculty in FLEX will report greater satisfaction with their clinical teaching experiences and greater perceptions of safety, teamwork and supervision than faculty in STD.*

*H2d: Standardised test scores for interns in Flex will not be less than (inferior to) those for interns in STD.*

**Specific aim 3:** Examine intern sleep time and alertness under STD and FLEX duty hour schedules.

### Secondary hypotheses
*H3a: Average daily sleep obtained by interns in FLEX will not be less than (will not be inferior to) that of interns in STD, as determined by a 14-day period of sleep monitoring using actigraphy and daily sleep diaries.*

*H3b: Interns in FLEX will not have (will not be inferior to) greater average subjective sleepiness via the Karolinska Sleepiness Scale (KSS) (24) or lower average behavioural alertness via the Psychomotor Vigilance Test (PVT) (25) than interns in STD, as determined by a 14-day period of morning sleepiness-alertness monitoring.*

ICOMPARE's sleep hypotheses are that the average daily sleep will not be less among interns in the FLEX arm compared with those in the STD arm and that interns in the FLEX arm will not have greater subjective sleepiness or lower behavioural alertness than interns in the STD arm.

## METHODS AND ANALYSIS
### Study design
iCOMPARE is a cluster-randomised trial to compare two alternative duty hour standards in 63 IM training programmes in USA fielded in academic year 2015–2016. Table 1 compares the duty hour standards between the FLEX and STD arms.

Cluster randomisation occurred at the level of the residency programme, the level at which duty hour policies

**Table 1**  Overview of usual care and intervention arm duty hour requirements

| Requirement | Usual care (STD; current duty hour standards) | Intervention (FLEX; flexible duty hour standards) |
| --- | --- | --- |
| 80 hours per week (averaged over 4 weeks) | No change | No change |
| 1 day off per week (averaged over 4 weeks) | No change | No change |
| In-house call no more than every third night (averaged over 4 weeks) | No change | No change |
| PGY-1 resident duty hour periods must not exceed 16 hours | No change | Eliminated |
| PGY-2 and above must not work more than 24 hours with an additional 4 hours permitted for transitions in care | No change | Eliminated |
| Residents must have 14 hours off after 24 hours in-house duty and at least 8–10 hours off after a regular shift | No change | Eliminated |

Adapted from Bilimoria *et al*.[27]
FLEX, flexible; PGY, postgraduate year; STD, standard.

are implemented. Although duty hour standards are mandated, individual programmes vary considerably in how they schedule their trainees within those standards. The trial is pragmatic in that the intervention arm effectively represents the exposure of residency programmes to an alternative set of duty hour standards. The exposure is the policy change, not the actual duty hours that are implemented in response. This approach is akin to clinical trials of outpatient pharmaceuticals in which the exposure is the prescription of the control or intervention drug, regardless of participant adherence.

Hypotheses regarding how interns spend their time require detailed time-motion observations, and hypotheses about interns' sleep and alertness require detailed observations of sleep cycles and psychomotor vigilance—both accomplished through substudies deployed in a sample of participating programmes.

### Sample size and power
The non-inferiority 30-day patient mortality hypothesis (H1a) is the trial's primary hypothesis. Defining the outcome as the trial year rate minus the pretrial year rate adjusts each programme's outcome for secular trends in 30-day mortality as well as for differential program-to-program

patient risk profiles. Using a two-sample t-test for non-inferiority of the between group mean year over year difference in 30-day mortality and assuming 80% power, Type I error of 5%, a non-inferiority margin of 1%, pooled SD for the outcome of 1.5% and a 30-day mortality rate of 11%, we calculated a required sample of 58 programmes, 29 per treatment group. The pooled SD of the outcome (year over year difference in 30-day mortality) and the 30-day mortality rate in the STD group were estimated using available Medicare claims from 2007 to 2008 for the population of target IM programmes (ie, all IM programmes meeting the trial's size and population criteria).

This number of programmes and the associated trainees and faculty provide excellent power for analyses investigating the education hypotheses which use person-level outcomes. Assuming 30 interns, 50 second year or higher trainees, 10 faculty members and 1 director per programme, iCOMPARE was projected to include 1740 interns, 2900 second year or higher trainees, 580 faculty members and 58 programme directors.

Hypothesis H2a addresses the Flex versus STD difference in per cent of time spent by interns in direct patient care and was assessed with a Time-Motion study. The sample size of the Time-Motion study is six programmes (3 FLEX, 3 STD; 60 interns, 10 interns per programme). Preliminary data[20 29] suggested that the mean per cent time spent in direct patient care in STD programmes would be 13%±4%. The Time-Motion study has at least 80% power to detect a 3% difference between FLEX and STD in time spent in direct care.

Hypothesis H3a addresses the FLEX versus STD difference in average intern daily sleep over a 14-day period. Targeting 90% power, one-sided Type I error of 5%, a non-inferiority margin of 0.5 hours and assuming expected average daily sleep in STD of 7±1.5 hours,[30] the sample size for the Sleep and Alertness study was calculated to be 290 interns (145 per treatment group) and increased to 384 interns (192 per treatment group) to anticipate data loss related to non-adherence and dropouts.

### Study population and inclusion criteria

The CONSORT diagram is provided in figure 1. In 2014, there were 379 IM ACGME-accredited training programmes in the country not on probation. The 54 programmes in New York State were excluded because that state's legislated duty hour standards were not subject to the waiver required for the intervention arm. Since the patient safety outcomes would be measured in Medicare patients, Veterans Administration hospitals did not qualify for inclusion. Sufficiently precise estimates of our patient safety outcome measures necessitated exclusion of 84 programmes training only at hospitals in the bottom 50% by trainee-to-bed ratio or the bottom 25% by patient volume in the measured diagnoses. In about 50% of teaching hospitals, the number of residents per bed is so few that the residents' impact on patient care is minimal, and so any changes in the impact of those residents' schedules would be too insignificant to measure.

An additional 62 programmes in the lowest quartile of number of trainees were excluded to ensure measurement of enough trainees. Two other programmes in the lowest quartile of number of trainees were allowed by the Steering Committee. One programme was approved for inclusion despite deviations in size of its two affiliated hospitals because together the two hospitals met the qualifying hospital size. A second programme was approved for inclusion because the hospital it was affiliated with had far exceeded the trainee-to-bed ratio even though the patient volume was slightly low. A total of 179 programmes were eligible for inclusion.

### Recruitment

Directors of eligible programmes were invited to participate by letters of invitation and informational sessions during the Fall 2014 and Spring 2015 Association of Program Directors in Internal Medicine (APDIM) meetings. Participation required agreement to support study needs and a participation agreement signed by the Designated Institutional Official indicating approval from the institution's Graduate Medical Education Committee. Programme directors indicated the hospitals, and the trainee rotations within those hospitals, which would participate in iCOMPARE. Each programme was reviewed by the study team to confirm that at least one of the hospitals indicated by the programme director qualified for inclusion in the trial. The study team checked the list of hospitals provided by the programme director for shared staffing with other training programmes eligible for and interested in iCOMPARE so that those programmes could be randomised together and avoid having different duty hour schedules for trainees at a common hospital. This occurred for 3 pairs of hospitals. In total, 63 programmes agreed to participate and were randomised.

### Patient and public involvement

The study was based on a fundamental concern about the impact of training schedules on patient, trainee and programme director experiences. However, no patients were directly involved in the design and conduct of the study. Programme directors from each participating programme were involved and essential to recruitment which was at the training programme level. Programme directors were fully informed about the intervention and individual trainees had the opportunity to decline participation in surveys and other education outcomes. Results will be presented at national meetings to programme directors and published in peer-reviewed journals.

### Institutional Review Board determinations

The University of Pennsylvania Institutional Review Board (IRB) approved the iCOMPARE protocol and served as the IRB of record for all participating programmes that agreed to sign an Institutional Affiliation Agreement. Forty programmes chose this option and submitted the required paperwork. Twenty-three programmes opted for a local review process.

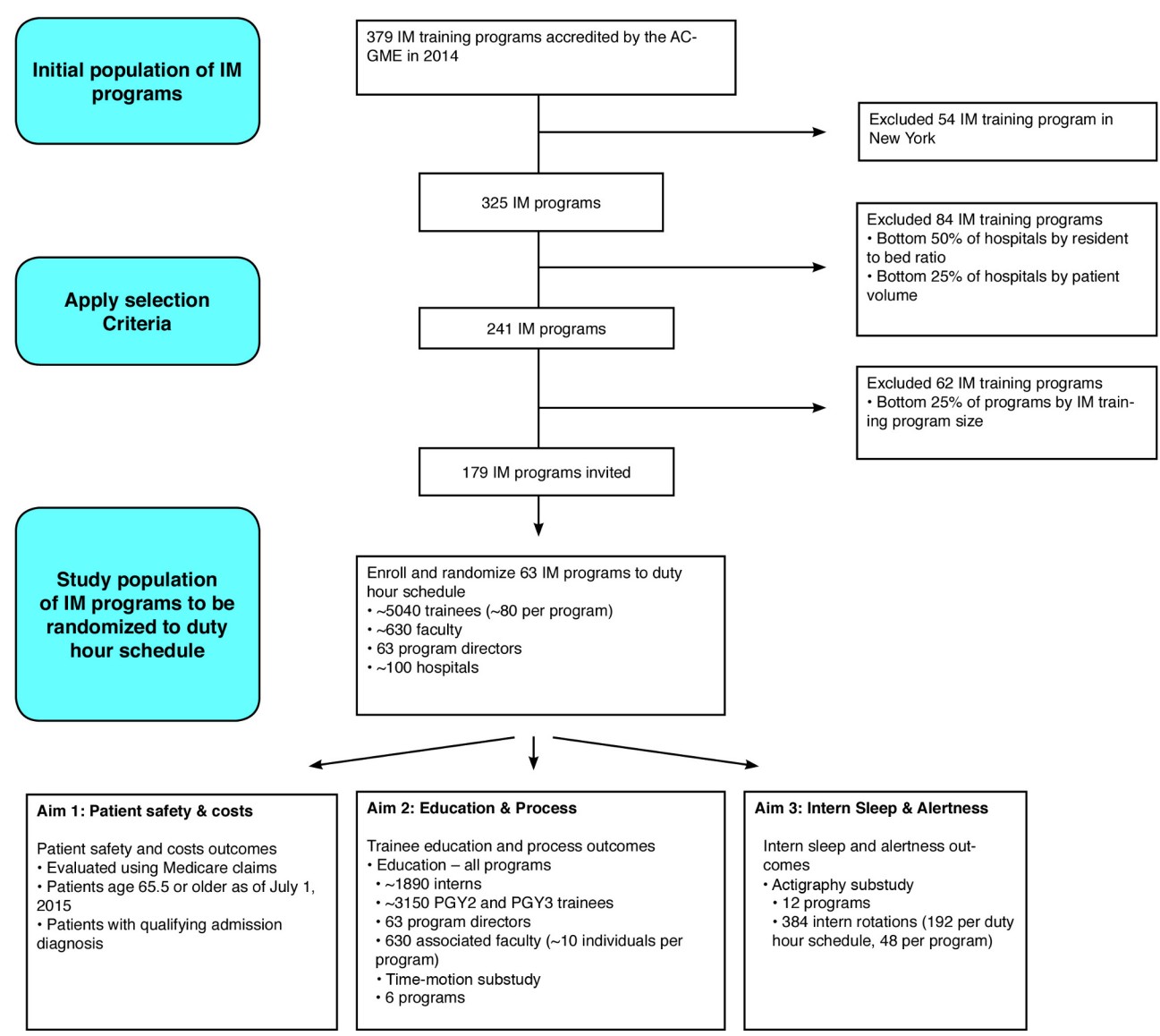

**Figure 1** Consort diagram. ACGME, Accreditation Council for Graduate Medical Education; IM, internal medicine.

## Randomisation

Randomisation was 1:1 into the STD and FLEX arms. Participating programmes in iCOMPARE were not masked to their assignment. Sixty programmes completed the application process quickly and were randomised in November 2014 so that they could provide applicants interviewing for positions in academic year 2015–2016 information about the duty hour arm assignment. Randomisation of programmes ended in March 2015, judged to be the latest date by which a programme could be randomised to the FLEX arm and have adequate time to adjust their residency schedules in anticipation of the 2015–2016 intervention year.

## Data sources

Hypothesis testing required outcome data from patients, trainees and programme directors and faculty. Because of the available and highly reliable patient mortality information (both in hospital and after discharge) in the Medicare fee-for-service (FFS) programme, the iCOMPARE

patient population was limited to Medicare FFS beneficiaries with a qualifying principal diagnosis on hospital admission (online supplementary appendix materials 1); these diagnoses were chosen for their common treatment on IM services (excluding oncology and neurology diagnoses) and their elevated mortality rates. Because similar data are not available for patients enrolled in a Medicare managed care programme, patients enrolled in a Medicare managed care programme 6 months before or 1-month postdischarge for the index hospital admission were excluded.

Randomised programmes were further invited to participate in the observational Time-Motion and the Sleep and Alertness substudies, described below.

## Timeline

Trial development began in 2013. The Medicare data required for the analyses of the patient safety outcomes are not expected to be available until mid 2018 (figure 2).

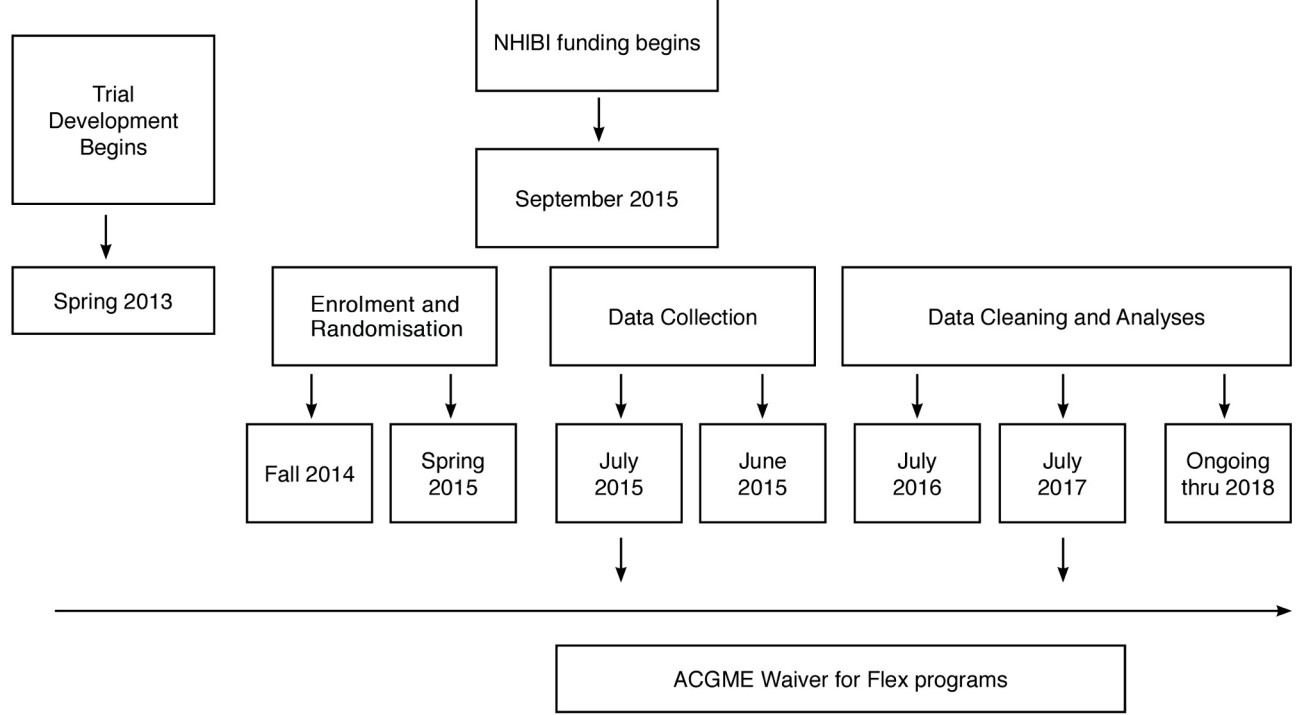

**Figure 2** iCOMPARE trial timeline. ACGME, Accreditation Council for Graduate Medical Education; iCOMPARE; individualized Comparative Effectiveness of Models Optimizing Patient Safety and Resident Education; NHLBI, National Heart, Lung, and Blood Institute.

## Outcomes, measures and data collection

### Aim 1: Patient safety

Patient safety measures are derived from Medicare data, collected as part of routine clinical practice and administration and obtained through ResDAC (https://www.resdac.org/). The primary outcome is based on 30-day (from admission) all-location mortality, a patient-centred measure unaffected by length of hospitalisation.[10 11 15 18 19 31 32] Secondary outcomes include prolonged length of stay,[14 33–35] 30-day readmission rate,[19 36] complication rate[13 33 37 38] and resource utilisation measures (cost)[17 34] and payment among FFS Medicare beneficiaries with specific diagnoses. Details regarding justification for and operationalisation of these outcomes are provided in online supplementary appendix materials 2.

### Aim 2: Education

Education measures are derived from multiple sources. The primary education measures are specified in table 2 and come from the Time-Motion substudy, the ACGME year-end trainee and core faculty surveys, and the interns' In-Training Examination (ITE) scores provided by the American College of Physicians (ACP). Secondary education measures include iCOMPARE-generated surveys and survey data provided by the APDIM.

### Time-motion substudy

We recruited three IM programmes randomised to STD and three randomised to FLEX to participate in direct observations of some of their interns, targeting programmes in the mid-Atlantic region for operational

convenience; programmes with tertiary hospitals as well as community-based programmes were included in both arms. Participating programmes received $3000 to support these substudy activities. We recruited interns rotating on General Medicine services between March-May 2016. Eligible and interested interns provided written consent. Among the 129 interns invited to participate, 120 (93%) consented.

Twenty-three observers (medical students and undergraduates) were trained to follow participating interns. They used a custom-built tablet-based software programme to document start and stop times for various intern activities: direct patient care, indirect patient care, education, rounds, work, handoffs and miscellaneous, each with various subcategories reflecting greater specificity of tasks. For example, direct patient care had subcategories for patient interactions, family interactions and physical contact (eg, physical examination). At least one activity had to be selected at all times, although more than one could be selected to reflect multitasking. At the start and stop of the shift an observer completed brief surveys that summarised total patient census numbers for that intern, including the numbers of transfers, discharges, admissions and patients received at the beginning of a shift and handed-off at the end.

Shifts were selected at each site aiming to capture 30 shifts in a proportion mirroring—specific for each site— how interns generally spend their time on a general medicine inpatient rotation in a given week. Observers shadowed approximately 8–10 shifts, for 1–3 shifts per

**Table 2** Operationalisation of primary education measures

| Hypothesis | Primary data source | Who | Definition |
|---|---|---|---|
| 2a: Time in direct patient care and education | Time-Motion substudy | PGY1 | Per cent of observed shift minutes spent in each activity |
| 2b: Trainee satisfaction | ACGME year-end trainee survey | PGY1–3 | A single item on the ACGME resident survey re: appropriate balance for education with response options of never, rarely, sometimes, often, very often |
| 2c: Programme director satisfaction | ACGME core faculty survey | Core faculty | A single item on the ACGME faculty survey that asked about residents'/fellows' workload exceeding the capacity to do the work with response options of never, rarely, sometimes, often, very often |
| 2d: Standardised test scores | ACP in-training examination scores | PGY1 | Per cent of questions answered correctly |

ACGME, Accreditation Council for Graduate Medical Education, ACP, American College of Physicians; PGY, postgraduate year.

intern. Extended shifts (eg, 24 hours call cycles) were often split by two observers. A 10% sample of shifts was observed simultaneously by two observers to estimate inter-rater reliability. The methods for the Time-Motion substudy were based on previous research in terms of the structure of observations (eg, categories used), process of observations (eg, platform used and scheduling strategies) and the training of observers.[27 28] Inter-rater reliability was high for pairs of observers using a training video (median kappa=0.67 and median agreement of 90%) as well as the real-time observations (median kappa of. 74 and median agreement of 89%). Interns received a $50 Amazon gift card for each observed shift.

### ACGME end-of-year survey data

The ACGME provided iCOMPARE with summary data derived from their end-of-year surveys of trainees and core faculty in May 2014, May 2015 and May 2016; all data provided by the ACGME to iCOMPARE was at the level of treatment group arm (FLEX or STD) and year collected (2014, 2015, 2016). All data were stripped of programme identifiers and individual respondent identifiers. iCOMPARE received summary treatment-group level information for the single question items specified

in hypotheses H2b and H2c and many content areas (combined responses over several questions covering related content) that comprise secondary education outcomes.

### ITE score data from ACP

The ACP provided iCOMPARE with trainee level ITE scores from 2015 and 2016 for those trainees who had given permission to share their score for research purposes. Identifying information for a score was limited to postgraduate year (PGY) year and programme identifier.

### APDIM annual survey data

The APDIM provided data from its survey of programme directors in fall 2015 and fall 2016. The deidentified data set did not include name of programme or programme director but did have a flag for STD or FLEX or not in study.

### End-of-shift surveys

From 31 August 2015 to 26 April 2016, iCOMPARE conducted 16 2-week cycles of daily surveys administered to all trainees at all participating programmes. At the start of each cycle, we randomised each trainee into one of 14 groups and then allocated each group to a day of the 2-week period to receive a survey. Thus, each trainee was surveyed once every 2 weeks.

Surveys alternated between two sets of questions. Survey 1 was sent every other day and asked: (1) the name of the rotation a trainee was on in the past 24 hours, specifying details such as inpatient or not, type of inpatient rotation, at main teaching hospital or other setting or not in hospital; and if the trainee was on an inpatient rotation; (2) number of new patient evaluations completed in past 24 hours; (3) number of handoffs experienced in past 24 hours (4) and the number of patients for which the trainee was the primary provider. These questions provide another view of how interns spend their time.

Survey 2 was sent on the alternate days and asked the same first question as Survey 1 as well as the trainee's ratings (too little, just right, too much) for (1) time spent in educational conference and related activities, (2) sense of ownership of patients, (3) work intensity and (4) continuity of care. Data for these questions are related to satisfaction and complement the iCOMPARE's end-of-year survey (described below) and ACGME survey (described above).

Trainees were entered into an incentive lottery designed so that in each 2 week cycle, one intern and one resident at each of the 63 IM programmes were each awarded either a $25 or $100 Amazon gift card if they had completed their survey during that period. After the first cycle, the cycle response rate ranged between 39% and 42%.

### iCOMPARE end-of-year surveys

All trainees and programme directors received an iCOMPARE study-specific survey in May–June 2015 (baseline) and again in May–June 2016 (postintervention),

administered via Lime Survey (https://www.limesurvey.org/). Invitations with personalised links were emailed during May of 2015 and May of 2016 with up to six reminders to non-respondents. At the end of the intervention year, a $2500 cash incentive was provided to each of the nine programmes with the highest response rates.

The trainee survey was administered to all trainees with only slight differences between versions for interns and PGY2 and higher trainees. The instrument was initially developed for the FIRST trial[26] and included items on trainee satisfaction, experience of duty hours, supervision, fatigue management and resident and patient safety and ended with the Maslach Burnout Inventory-Human Services Survey (MBI-HSS).[39] The MBI-HSS is a 22-item rating scale assessing three domains: emotional exhaustion (nine items), depersonalisation (five items) and lack of personal accomplishment (eight items). Items are answered on a frequency scale of 0 to 6 where 0 indicates never and 6 indicates every day.

The programme director survey was modelled from an earlier survey to programme directors[40] and included items on resident and faculty workload, resident morale, continuity, education, patient safety and programme finances and administration.

### Aim 3: Sleep and alertness

Outcomes for the third aim include sleep duration and both subjective and objective measures of alertness among interns at six sites randomised to STD and six sites randomised to FLEX. At each of these 12 sites, programme coordinators recruited interns scheduled to be on general medicine, medical intensive care, cardiology or cardiac care rotations. Each participating programme received $8000 to cover the costs associated with scheduling interns for data acquisition and managing study equipment. Data collection spanned 5 November 2015 to 31 May 2016. The 12 programmes in the Sleep and Alertness Substudy were selected from across the country and represented programmes of various sizes. A total of 457 interns at these 12 programmes participated in the consent process; 432 (94.5%) provided written consent, 12 of whom were not eligible due to rotation schedules. Of the remaining 420, 398 (94.8%) provided data for the study and 22 (5.3%) discontinued participation.

Participating interns underwent 14 days of continuous measurements of rest and activity via wrist actigraphs (model wGT3X-BT, The Actigraph, Pensacola, Florida, USA).[41–43] Interns were instructed to wear the actigraph continuously, even on days off, except during activities that might damage it (eg, water immersion or contact sports) or that would impede the delivery of clinical care. They were asked to remove the actigraph for up to 2 waking hours for recharging on days 1 and 7. Each participating intern received a $10 Amazon gift card for each day for which data were received.

Each morning, the intern completed a brief online survey including the name of the shift that the intern was working; a log reflecting sleep times, sleep quality and experiences of excessive sleepiness and the Karolinska Sleepiness Scale.[44] The interns then completed a 3 min Psychomotor Vigilance Test (PVT-B).[45 46] All surveys and assessments were completed on a study-supplied Android Smartphone (Samsung Galaxy SIII Neo). The PVT-B is based on simple reaction time to stimuli occurring at random interstimulus intervals and is the gold standard for measuring the neurobehavioral effects of acute and chronic sleep loss and circadian misalignment.[47] Actigraphy, survey and PVT-B data were automatically encrypted and uploaded to a secure server and checked daily for protocol adherence and potential technical issues with the equipment. If problems were detected or no data were received, interns were contacted by the study team to resolve any difficulties interns were having with equipment. Details of scoring are presented in the online supplementary appendix materials 3: Sleep Actigraphy Scoring and online supplementary appendix figures 1 and 2.

### Statistical considerations

Non-inferiority tests will be one-sided and superiority tests will be two-sided. All primary analyses will compare the FLEX and STD treatment groups as randomised, regardless of adherence to the assigned duty hour standards, according to the intention-to-treat principle. Since directors at programmes assigned to FLEX have considerable latitude in design of trainee schedules, we expect variation among the duty hour schedules followed in the FLEX group. Protocol-specified secondary analyses addressing the degree of difference between FLEX and STD schedules will be completed. We will also report mortality results adjusting for the clinical condition associated with patient's principal diagnosis as well as demographic variables and comorbidities determined using a 6-month look-back period. Similar approaches will be used for the other patient safety outcomes specified in hypotheses H1b–H1e.

The outcome measures for the education hypotheses (H2a–H2d) and the sleep and alertness hypotheses (H3a–H3b) are person-level (intern, trainee or faculty) measures. If we have repeated measures of an outcome on an individual (eg, an intern has observations of time spent in direct patient care over multiple shifts, an intern has observations of time spend sleeping over more than 1 day), we will analyse the mean of the repeated measures on the individual, which eliminates within-person correlations. Each outcome will be the response variable in a mixed effects linear or logistic regression model with a random intercept for each programme cluster and a single fixed term for treatment group as implemented in SAS (Cary, North Carolina, USA) or STATA (College Station, Texas, USA). When pretrial year data are available, additional analyses will adjust an individual's trial year mean response for the pretrial year programme-level mean response (ie, a difference of differences analysis). The pretrial programme level mean response must be used since the trainees and faculty providing data in

the pretrial year are not the same trainees and faculty providing data in the trial year, but are at the same IM programme. Specifically we have pretrial data for Medicare patient level safety analyses and for some Education outcomes (the ACGME end-of-year surveys, ITE score data from ACP and the iCOMPARE end-of-year surveys).

## DISSEMINATION

Plans for dissemination include submitting results for the various aims to academic meetings and peer-reviewed publications as they become available. The Data and Safety Monitoring Board will comment on manuscripts reporting results related to the primary hypotheses before journal submission.

## DISCUSSION

iCOMPARE is a 1-year, cluster-randomised, pragmatic trial designed to evaluate the availability of an alternative resident duty hour schedule for effects on patient safety, resident education and intern sleep and alertness. iCOMPARE's design is similar to that of the FIRST trial conducted among surgical trainees. What distinguishes both of these trials from prior work investigating effects of duty hour schedules, in addition to their large size, comprehensive approach and immediate policy relevance, is that they help elevate the standards of evidence applied to graduate medical education policy more generally in USA and abroad. Although randomised trials provide strong evidence for the causal inference required for good policy, they typically answer narrow questions. Those questions today revolve around the length of shifts within an 80 hours work week. Inside the field of graduate medical education, these issues have been hotly debated, although the 80 hours in USA far exceed the limits set for other counties with similar educational systems.[3 4]

iCOMPARE has some limitations. The measurement of patient safety relies predominantly on mortality effects, an important but incomplete measure of relevant clinical outcomes. The measurement of education relies considerably on existing and conventional tests and surveys and those developed specifically for this study, whereas education is more dimensionalised and, indeed, its own value is perhaps better reflected in changes in how well physicians perform through their careers, unmeasured in this study. Nevertheless, this pragmatic study of healthcare policy targets measures that are likely to be influential in policy decisions about resident duty hours and available in a time frame when the answers to those questions remain relevant.

Some may wonder why duty hours or shift lengths were not imposed on the intervention group and why, instead, the intervention group merely had permission to use more flexibility in their scheduling. The design of this study means that the potency of the duty hour changes actually implemented by programmes may be less extreme than what was permitted by the intervention, adding noise

to or potentially weakening the overall observed effect. Those concerns are valid to the extent the question is: What happens to patient safety, education or sleep and alertness when residents are forced to work in certain shifts? But the question at hand is: What happens when programmes are allowed flexibility in their scheduling of shifts? The outcome of those policy decisions is, in its implemented state, a product of the flexibility of the rules and the degree to which individual programmes take advantage of that flexibility. In some ways, this study design is consistent with that of effectiveness trials of drugs, where the anticipated effect is a product not just of whether one was randomised to the study drug, but also whether one was adherent to the drug. In the real world, adherence is relevant. Context is similarly relevant. The effects we observe in this trial also depend critically on the oversight and supervision provided to interns by more senior residents and on other safety nets built into the environment of hospital practice. While those safety nets potentially blunt observed effects, they take this study beyond the in vitro relevance of laboratory study to a pragmatic context.

With the results of the FIRST trial demonstrating non-inferiority in patient outcomes when a more flexible schedule was available, the ACGME issued new duty hour standards, effective July 2018, that correspond to the intervention arm of iCOMPARE.[48]

Given that surgery and IM are large fields with many residents caring for many patients, it is important to study the duty hour rules in both specialties as surgical and medical training programmes differ in structure, process, culture, the kinds of residents they attract, the patients they serve and the duties of trainees.

## CONCLUSION

There is considerable interest in understanding concerns in the US healthcare system, such as why US healthcare is so expensive, why its overall effects on health lag behind those of peer nations in particular with respect to safety, and why its individual effects are so unevenly distributed. In contrast, relatively little effort has been devoted to studying how physicians are trained—surprising because although physicians are certainly not the only reason for the concerns, it seems evident that physician training must play an important role in most outcomes related to US healthcare. In that broad context, iCOMPARE promises narrow but substantive evidence to inform resident duty hour standards in IM and signals new interest in methodologically stronger research in medical education.

**Author affiliations**
[1]Department of Medicine, University of Pennsylvania, Philadelphia, Pennsylvania, USA
[2]Department of Pediatrics, Children's Hospital of Philadelphia, Philadelphia, Pennsylvania, USA
[3]Department of Medicine, The Johns Hopkins University, Baltimore, Maryland, USA
[4]Department of Psychiatry, University of Pennsylvania, Philadelphia, Pennsylvania, USA

⁵Department of Biostatistics, The Johns Hopkins University, Baltimore, Maryland, USA

⁶Department of Epidemiology, The Johns Hopkins University, Baltimore, Maryland, USA

⁷Wharton Statistics Department, University of Pennsylvania, Philadelphia, Pennsylvania, USA

⁸Department of Medicine, Brigham and Women's Hospital, Boston, Massachusetts, USA

⁹Corporal Michael J. Crescenz VA Medical Center, Philadelphia, Pennsylvania, USA

¹⁰Department of Medical Ethics and Policy, University of Pennsylvania, Philadelphia, Pennsylvania, USA

**Acknowledgements** The iCOMPARE team extends their appreciation to the participating Program Directors, Jeremy M Asch, BA, Amanda K Bertram, MS, Michele M Carlin BA, Sara C Coats, BS, Adrian J Ecker, Kelsey A Gangemi, MPH, Alexander S Hill, BS, Susan K Malone, PhD, RN, Alyssa M Yeager, MD and Pulsar Informatics for their support of the trial.

**Collaborators** iCOMPARE Research Group.

**Contributors** JAS is a coinvestigator. She took the lead in drafting this paper and incorporating all coauthors' suggestions on subsequent drafts. She also worked closely with SVD and LMB to design and refine the aims and methods related to the education aim. JHS took the lead on writing the protocol for the patient safety component of the protocol. He reviewed and commented on multiple versions of this manuscript. SVD worked with JAS and LMB to develop the details for the education aim and outcomes. He reviewed and commented on multiple versions of this manuscript. DFD took the lead and worked with MB to develop the details for the sleep and alertness aim and outcomes. He reviewed and commented on multiple versions of this manuscript. LMB worked with JAS and SVD to develop the details for the education aim and outcomes. She reviewed and commented on multiple versions of this manuscript. JT took the lead and worked with ALS, DSS and DMS to develop the sample size calculations and analytic plan for all study aims. They each reviewed and commented on multiple versions of this manuscript. JTK participated in design of the direct observation portion of the protocol and commented on multiple versions of the manuscript. MB worked with DFD to develop the details for the sleep and alertness aim and outcomes. He reviewed and commented on multiple versions of this manuscript. KHC took the lead in developing the details for the direct observation portion of the protocol. He reviewed and commented on multiple versions of this manuscript. OE-S worked with JHS to design the patient safety portion of the protocol. DWB participated in design of the direct observation portion of the protocol and commented on multiple versions of the manuscript. KGV worked on the overall development of the protocol and commented on versions of the manuscript. DAA is the principal investigator for the iCOMPARE protocol and has been instrumental in all phases of protocol design and study implementation. The ICOMPARE Research Group, comprising all participating programme directors and project staff, was instrumental in recruitment and data collection.

**Funding** The trial was funded by grants from the National Heart, Lung, and Blood Institute (NHLBI); grant number U01HL125388 to Dr Asch and grant number U01HL126088 to Dr Tonascia) and the Accreditation Council for Graduate Medical Education.

**Competing interests** None declared.

**Patient consent** Not required.

**Ethics approval** University of Pennsylvania IRB, USA.

**Provenance and peer review** Not commissioned; externally peer reviewed.

**Author note** This paper is subject to NIH Public Access Policy.

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
