## [Reviewer comments · BMJ Open]

This paper was submitted to a another journal from BMJ but declined for publication following peer review. The authors addressed the reviewers' comments and submitted the revised paper to BMJ Open. The paper was subsequently accepted for publication at BMJ Open.

(This paper received three reviews from its previous journal but only two reviewers agreed to published their review.)

ARTICLE DETAILS

TITLE (PROVISIONAL)	Development of the Individualized Comparative Effectiveness of Models Optimizing Patient Safety and Resident Education (iCOMPARE) Trial: A Protocol Summary of a National Cluster-Randomized Trial of Resident Duty Hour Policies in Internal Medicine
AUTHORS	Shea, Judy A; Silber, JH; Desai, Sanjay; Dinges, David F; Bellini, Lisa M; Tonascia, James; Sternberg, Alice L; Small, Dylan S; Shade, David M; Katz, Joel Thorp; Basner, Mathias; Chaiyachati, Krisda H; Even-Shoshan, Orit; Bates, David Westfall; Volpp, Kevin G; Asch, David A; Research Group, iCOMPARE

VERSION 1 – REVIEW

REVIEWER	Lillian Emlet University of Pittsburgh Medical Center United States
REVIEW RETURNED	04-Mar-2018

GENERAL COMMENTS	The protocol for iCOMPARE is clearly delineated and thoughtfully executed with the clear goals of non-inferiority between the standard and flexible work schedules for Internal Medicine trainees. Main outcomes are 30-day mortality, Time Motion study (time spent with patients, sleep amount), educational impact (in training exam scores). Some assessments that are probably too global to be different include the ACGME end of year surveys or mortality/ readmission (other factors may impact more strongly). What will be interesting to see is if completion rate for every other day survey of subjects become too onerous for completion, if additional team members added to care teams (ie. NPs PAs) make any impact on work load/ hours, and if VA hospital exposure are equal between training sites. It is also unclear if faculty modifications to team ratios are accounted for. Overall a well done study protocol, and it will be interesting to see the training program required for student direct observers of the Time Motion Study in terms of inter-rater reliability and reproducibility of assessments.
--

REVIEWER	Daniel Furnivall Imperial College London, UK
REVIEW RETURNED	26-Mar-2018

GENERAL COMMENTS	This looks like a strong study, and I am happy to recommend publication. Small issues: There were several small english errors in the "strengths + limitations" and "author contributions" sections. Additionally, I was unable to locate the program director survey referred to on page 17 - although I may have misread this description. If this was non intended to be read by reviewers this may warrant rewording. Suggestions: It would be helpful if the authors could include more granular explanations of exactly what costs will be extracted - for example, how will costs of prescription or other medical errors be accounted for within the study? Are clinical negligence lawsuits in any way accounted for? Also, study limitations should be more extensively described towards the end of the paper. It seems the only limitation discussed here currently is the narrow focus.
--

REVIEWER	Anne Linker, MD Division of Hospital Medicine Icahn School of Medicine, Mount Sinai Hospital New York, NY USA
REVIEW RETURNED	05-Apr-2018

GENERAL COMMENTS	Summary Opinion & Recommendation: I recommend that this manuscript describing the protocol for iCOMPARE be accepted with plan for minor revisions. The protocol is well-described, but the authors could include a bit more information regarding the limitations of the protocol as drafted, data to support the methods for the time-motion study, and information regarding the geographic distribution of programs included in the time-motion study and sleep/alertness analysis. It would also be helpful to include some description of comparative analyses were not made for trainees and faculty comparing the intervention year to other years, as this comparison may have further clarified the benefits or drawbacks of the intervention. Comments: The authors clearly define the study objectives and the hypotheses set forth prior to the start of the study. The abstract is complete, and appears to be balanced. Furthermore, the pragmatic design is appropriate to answer the research question. Given that policy decisions from the ACGME are usually quite general and residency programs use different methods to achieve compliance, it is appropriate to study the outcomes discussed under conditions that simulate a policy change, rather than by studying specific work schedules. However, the authors only briefly acknowledge that differences in approach towards compliance (different schedules) may cause the study effect to be diluted or may present a challenge to identifying causative factors in effects on the outcomes of interest. The authors might also acknowledge a few other methodologic weaknesses to the study. First, because the programs opted in or out of the study, there is a selection bias for all involved programs. The trial is not blinded. Given the intervention under analysis, this is understandable. A more significant weakness is present in the time-motion study. Interns opted in to the time-motion study, and there is little discussion of what percent of interns at the sites queried opted in. Given that the outcomes from the time-motion
--

	and alertness parts of the study are extremely important to educators and others reading the study, it seems paramount to explain how valid these results are by further describing the study group and any bias that may have come into play for the population included. It is also unclear why the time-motion study is conducted only for interns, and did not include residents. Interns have the greatest number of fixed tasks compared to other trainees. It would be interesting to know if educational time or time with patients increased for residents in the intervention arm, as this would add to the literature regarding benefits of flexible schedules. Critical readers would benefit from a discussion of this decision. In addition, data is collected for the study during only one calendar year. This seems to me to be a challenge in the study, though may have been unavoidable depending on study design, funding, etc. It often takes some time for residents and faculty to acclimate to new changes in a residency program. In particular, the residents in the program are training in an era in which long work hours are often criticized. It might have been interesting to include some pre-intervention data on preconceptions of long work hours from trainees, faculty, and program directors. In addition, because data was collected for only one year, it's not clear if the trainees had the opportunity to compare across different schedules. It is not clear if questions comparing the schedules were asked of the upper level trainees, who would have experienced both restricted and flexible duty hours. Furthermore, it would have been interesting to note if the educational outcomes were different in the year prior to the intervention, instead of only comparing across the two groups. (For instance, if intraining exam scores improved from before the intervention to after the intervention, etc.) One major criticism of the FIRST trial was that it did not account for the effect of oversight on trainees. In other words, if there is adequate oversight one would not expect (or at least one would hope not to find) a significant difference in patient mortality and/or safety outcomes due to changes in resident schedule. The authors could address this concern more directly, as it is likely to arise for the iCOMPARE trial as well.
--	---

REVIEWER	Pedro Ramos Karolinska Institutet, Sweden
REVIEW RETURNED	16-Apr-2018

GENERAL COMMENTS	Comments to the Authors Report on "Development of the Individualized Comparative Effectiveness of Models Optimizing Patient Safety and Resident Education (iCOMPARE) Trial: A Protocol Summary of a National Cluster-Randomized Trial of Resident Duty Hour Policies in Internal Medicine" This paper presents the study protocol for a cluster randomized trial designed to assess the impact of duty hour rules of internal medicine residency programs on patient safety, resident education, and intern sleep and alertness. According to figure 2, the trial is still ongoing, and outcomes related to patient safety are not available before mid-2018.
---

Findings are not reported.

General Comments:

1. The protocol is suitable for publication in BMJ Open, provided some minor details are incorporated in the manuscript. The study is innovative and very comprehensive; the focus of the trial is extremely relevant from a health policy standpoint and for both its patient safety and health economics dimensions and may be useful for ongoing discussions on medical education in the US and elsewhere. The study protocol is very well designed.

Specific Comments:

1. Introduction: I think the authors may motivate the trial a little bit better. The problem is only addressed from a US perspective. Has this problem been studied elsewhere? If not or if you aim to look at it from an entirely US perspective, you may consider writing a subsection that addresses the US Medical education context and helps non-US readers understand the system better (if they feel they need to).

2. Sample Size and Power: Please provide reference for the 30day mortality rate of 11.5% (pag.10) for calculating sample size.

3. Study population and inclusion criteria:

In pag. 11, you mention: "Sufficiently precise estimates of our patient safety outcome measures necessitated exclusion of 84 programs training only at hospitals in the bottom 50% by trainee-to-bed ratio or the bottom 25% by patient volume in the measured diagnoses".

The reason for excluding the programs with the lowest resident-to-bed ratio should be better explained. It seems that these programs would be the ones that could be impacted the most with changes in duty hours.

4. Data Sources:

In pag. 12, you mention: "these diagnoses were chosen for their common treatment on internal medicine services (excluding oncology and neurology diagnoses) and their elevated mortality rates"; it could be useful to elaborate more on the criteria for the choice of the ICD9. For instance, in cardiac disease, you chose 3 specific ICD codes; why were these 3 chosen and not others. This detail could be included in the annex, and not in the main document.

5. Outcomes , Measures and Data Collection:

In Aim 1 – Patient Safety, outcome measures are insufficiently described. For easiness of interpretation, you could detail the measures studied and how you collected them. You mentioned before that mortality data is reliable; is complication rate data as reliable? You used complication rates defined by AHRQ Patient Safety Indicators? Which are these? Were there any process

	guidelines for the programs on how to measure each or are these sufficiently standardized? 6. Discussion: Are there any limitations you were willing to accept a priori ? These could be described. 7. Other sections: No additional comments.
--	---

VERSION 1 – AUTHOR RESPONSE

1. Some assessments that are probably too global to be different include the ACGME end of year surveys or mortality/ readmission (other factors may impact more strongly).

We agree that some of the outcomes measures we chose were global, but they are well recognized metrics by program directors in the United States. In fact, in our initial publication there are no differences in the ACGME surveys. Data for mortality/readmission are yet to be done. These outcomes are standard for assessing duty hour impacts.

1Desai SV, Asch DA, Bellini LM, Chaiyachati KH, Liu M, Sternberg AL, Tonascia J, Yeager AM, Asch JM, Katz JT, Basner M, Bates DW, Bilimoria KY, Dinges DF, Even-Shoshan O, Shade DM, Silber JH, Small DS, Volpp KG, Shea JA for the iCOMPARE Research Group. Education outcomes from a Duty-Hour flexibility trial in Internal Medicine. N Engl J Med 2018; 378:1494-1508. DOI: 10.1056/NEJMoa1800965

2. What will be interesting to see is if completion rate for every other day survey of subjects become too onerous for completion,

After the first cycle, the cycle response rate ranged between 39% and 42%. This sentence was added to page 17.

3. ..if additional team members added to care teams (ie. NPs PAs) make any impact on work load/ hours, and if VA hospital exposure are equal between training sites.

This is an interesting idea but we did not look at change to care teams or include VAs.

4. It is also unclear if faculty modifications to team ratios are accounted for.

This is another interesting change to consider. Unfortunately we did not collect these data.

5. it will be interesting to see the training program required for student direct observers of the Time Motion Study in terms of inter-rater reliability and reproducibility of assessments.

Inter-rater reliability for pairs of observers using a training video was high for recording the same activity at the exact second, within three seconds, and within five seconds (median kappa of 0.66, 0.67 and 0.71, and median agreement of 90%, 90%, and 91%, respectively). In addition, two observers were deliberately overlapped for 10% of shifts during which inter-rater reliability remained high (median kappa of .74 and median agreement of 89%). A summary of this information was added to page 15.

6. Small issues: There were several small english errors in the "strengths + limitations" and "author contributions" sections.

Thank you for the careful review. We believe all errors have been corrected.

7. Additionally, I was unable to locate the program director survey referred to on page 17 - although I may have misread this description. If this was non intended to be read by reviewers this may warrant rewording.

Thank you – this reference to an online example was outdated and removed.

8. Suggestions: It would be helpful if the authors could include more granular explanations of exactly what costs will be extracted - for example, how will costs of prescription or other medical errors be accounted for within the study? Are clinical negligence lawsuits in any way accounted for?

This detail has been added to Supplementary Appendix: Appendix Materials 2. (pages 14, 16-17)

Costs are calculated using a resource utilization-based method of cost estimation. The following items, which are calculated using the Medicare claims data, are included in the total cost estimate:

- Accommodation costs, which are based on the number of general floor days and the number of intensive care unit days during the index admission. This information comes from the revenue center files.*
- Operating room cost, based on the amount of time spent in the operating room (for patients who had a surgical procedure performed). This is determined using Part B claims.*
- Emergency room visit fixed costs, based on post-discharge visits to the emergency room. This is determined using Part B claims.*
- Costs of services provided, based on Relative Value Units (RVUs), determined using the Current Procedural Terminology codes on bills. This is determined using Part B claims.*

In addition, any costs that occurred within 30 days of the index admission date, and all the costs associated with any readmissions that began within 30 days, are also included in the total cost calculation.

Drugs administered during the hospitalization are included in the cost calculation, but drugs outside the hospital are not. Costs of lawsuits are not available in the Medicare claims data.

Hospital medical errors that resulted in a longer length of stay or the use of additional hospital resources, such as the need to admit a patient to the intensive care unit, would be captured in the cost calculation.

9. Also, study limitations should be more extensively described towards the end of the paper. It seems the only limitation discussed here currently is the narrow focus.

We have added two paragraphs (pages 21-22) of limitations.

10. The protocol is well-described, but the authors could include a bit more information regarding the limitations of the protocol as drafted,

See response to #11 – two paragraphs have been added.

11. data to support the methods for the time-motion study,

A summary of this detail was added to page 15.

The methods for the time-motion study were based on previously published research in this type of study. One of our investigators participated in one of these previous studies and we discussed the

methods with other investigators of previous research. The specific studies used to inform our methods are listed below. The previous research was used to develop the structure of observations (e.g. categories used), process of observations (e.g. platform used and scheduling strategies), and the training of observers.

1. Block L, et al. In the wake of the 2003 and 2011 duty hours regulations, how do internal medicine interns spend their time? *J Gen Intern Med* 2013;28:1042-7.

2. Fletcher KE, et al. The composition of intern work while on call. *J Gen Intern Med* 2012;27:1432-7.

12. information regarding the geographic distribution of programs included in the time-motion study and sleep/alertness analysis.

We have added sentences to pages 14 and 18. As noted on page 14, programs participating in the time motion study were drawn from the mid-Atlantic region for feasibility, but they included programs with tertiary hospitals as well as community based programs in both arms.

The 12 programs in the sleep/alertness study were selected from across the country and represented programs of various sizes. (page 18)

13. It would also be helpful to include some description of comparative analyses were not made for trainees and faculty comparing the intervention year to other years, as this comparison may have further clarified the benefits or drawbacks of the intervention.

We agree and think this is a strength of our design. We have clarified (page 20) that we did use pretrial data in some cases. Specifically we have it for the Medicare patient level safety analyses, and the Education outcomes including ACGME end of year surveys, ITE score data from ACP, and the iCOMPARE end-of-year surveys, ACGME surveys and ACP scores.

14. However, the authors only briefly acknowledge that differences in approach towards compliance (different schedules) may cause the study effect to be diluted or may present a challenge to identifying causative factors in effects on the outcomes of interest.

The extended discussion on pages 21 and 22 addresses this point.

15. The authors might also acknowledge a few other methodologic weaknesses to the study. First, because the programs opted in or out of the study, there is a selection bias for all involved programs.

See our responses to #11 – Indeed, the selection bias was noted in the recent NEJM iCOMPARE paper.¹

16. A more significant weakness is present in the time-motion study. Interns opted in to the time-motion study, and there is little discussion of what percent of interns at the sites queried opted in.

Among the 6 programs that participated in the time-motion study there were 237 interns. Of these 108 were not approached/invited. Among the 129 interns invited to participate, 120 (93%) consented. (added to page 14)

17. Given that the outcomes from the time-motion and alertness parts of the study are extremely important to educators and others reading the study, it seems paramount to explain how valid these results are by further describing the study group and any bias that may have come into play for the population included.

See response to previous comment. For time-motion, Among the 129 interns invited to participate, 120 (93%) consented.

Among the 12 programs participating in the Sleep-Alertness substudy, 457 interns participated in the consent process; 94.5% (n=432) provided written consent. Of the 432, 12 were not eligible due to rotation schedules. Of the remaining 420, 398 (94.8%) provided data for the study; 5.3% discontinued participation. (added to page 18).

18. It is also unclear why the time-motion study is conducted only for interns, and did not include residents. Interns have the greatest number of fixed tasks compared to other trainees. It would be interesting to know if educational time or time with patients increased for residents in the intervention arm, as this would add to the literature regarding benefits of flexible schedules. Critical readers would benefit from a discussion of this decision.

It is true that the intervention could have effects throughout the training hierarchy, but the direct effects were most directly relevant to interns. For budgetary and logistic reasons, we did not examine the time-motion effects on more senior residents.

19. In addition, data is collected for the study during only one calendar year. This seems to me to be a challenge in the study, though may have been unavoidable depending on study design, funding, etc. It often takes some time for residents and faculty to acclimate to new changes in a residency program. In particular, the residents in the program are training in an era in which long work hours are often criticized.

We agree that a longer time would have been preferable. In fact our original design called for a two year intervention with a cross-over design. Funders and stakeholders wanted a smaller study with a shorter timeline to results.

20. It might have been interesting to include some pre-intervention data on preconceptions of long work hours from trainees, faculty, and program directors. In addition, because data was collected for only one year, it's not clear if the trainees had the opportunity to compare across different schedules.

See our response to #15. We did use pretrial data. Interns could compare across different schedules in that flexible schedules were only implemented on some rotations, usually general medicine and the medical intensive care unit.

21. It is not clear if questions comparing the schedules were asked of the upper level trainees, who would have experienced both restricted and flexible duty hours.

No comparative questions were asked but upper level trainees were invited to complete all surveys.

22. Furthermore, it would have been interesting to note if the educational outcomes were different in the year prior to the intervention, instead of only comparing across the two groups. (For instance, if intraining exam scores improved from before the intervention to after the intervention, etc.)

See responses to #15 and #22. We did analyses controlling for pre-trial experiences. 1

23. One major criticism of the FIRST trial was that it did not account for the effect of oversight on trainees. In other words, if there is adequate oversight one would not expect (or at least one would hope not to find) a significant difference in patient mortality and/or safety outcomes due to changes in resident schedule. The authors could address this concern more directly, as it is likely to arise for the iCOMPARE trial as well.

This was addressed in the addition to the limitations (pages 21 and 22)

24. Introduction: I think the authors may motivate the trial a little bit better. The problem is only addressed from a US perspective. Has this problem been studied elsewhere? If not or if you aim to look at it from an entirely US perspective, you may consider writing a sub-section that addresses the US Medical education context and helps non-US readers understand the system better (if they feel they need to).

We have edited the abstract (p 4), the introduction (p 6) and the discussion (p 20, 21) to acknowledge duty hours are an international concern, especially among European countries and Canada. Given space concerns we do not go into great detail.

25. Sample Size and Power: Please provide reference for the 30day mortality rate of 11.5% (pag.10) for calculating sample size.

The 30-day mortality rate in the iCOMPARE target population was estimated to be 11% based on available Medicare claims from 2007-2008. We have added this detail to page 10.

26. Study population and inclusion criteria:

In pag. 11, you mention: "Sufficiently precise estimates of our patient safety outcome measures necessitated exclusion of 84 programs training only at hospitals in the bottom 50% by trainee-to-bed ratio or the bottom 25% by patient volume in the measured diagnoses".

The reason for excluding the programs with the lowest resident-to-bed ratio should be better explained. It seems that these programs would be the ones that could be impacted the most with changes in duty hours.

Our goal in excluding the hospitals with the lowest resident-to-bed ratios was to avoid including the "very minor" teaching hospitals (defined as those with an RB ratio less than or equal to 0.05) while still allowing the "minor" teaching hospitals (defined as those with an RB ratio between 0.05 and 0.25) to qualify for inclusion. In about 50% of teaching hospitals, the number of residents per bed is so few that the residents' impact on patient care is minimal, and so any changes in the impact of those residents' schedules would be too insignificant to measure.

The justification in the final sentence was added to page 11.

36. Data Sources:

In pag. 12, you mention: "these diagnoses were chosen for their common treatment on internal medicine services (excluding oncology and neurology diagnoses) and their elevated mortality rates"; it could be useful to elaborate more on the criteria for the choice of the ICD9. For instance, in cardiac disease, you chose 3 specific ICD codes; why were these 3 chosen and not others. This detail could be included in the annex, and not in the main document.

The explanation and justification for the selected codes is presented below and added to Supplementary Appendix: Appendix Materials 1 (page 2).

The original ICD-9 code list was created by searching for relevant codes for each medical condition that were found to be associated with high mortality rates in our preliminary 2008 data. The outcomes team reviewed the code list and expanded the code list in two ways:

- We consulted the official ICD-9 code book and looked for additional high-volume codes that were related to the medical conditions of interest. The list of proposed expansion codes was reviewed and approved by the coding consultant to the study, Dr. Patrick Romano of the University of California, Davis.
- To account for the presence of ICD-10 codes on claims with discharge dates beginning October 1, 2015, we utilized the General Equivalence Mappings (GEMs) made available by the Center of Medicare and Medicaid Services, which provide the closest possible approximation of a translation between the ICD-9 and ICD-10 code systems. Our goal was to translate each ICD-10 code back to its most closely equivalent ICD-9 code, and our review of the crosswalk caused us to pick up a small number of additional ICD-9 codes. As with the original code list expansion, these proposed additional codes were reviewed and approved by our coding consultant to the study.

27. Outcomes , Measures and Data Collection:

In Aim 1 – Patient Safety, outcome measures are insufficiently described. For easiness of interpretation, you could detail the measures studied and how you collected them. You mentioned before that mortality data is reliable; is complication rate data as reliable? You used complication rates defined by AHRQ Patient Safety Indicators? Which are these? Were there any process guidelines for the programs on how to measure each or are these sufficiently standardized?

A detailed explanation of the choices for outcomes is provided in Supplementary Appendix: Appendix Materials 2 (pages 13-17) and reproduced below.

All patient safety measures are based on information made available to researchers by the Center of Medicare and Medicaid Services (CMS). Health services researchers have conducted research using Medicare claims files for over two decades. This information is uniformly recorded for all Medicare beneficiaries. CMS releases to researchers the de-identified claim files for hospitals' and physicians' services after validating them through an adjudicated process. All the patient safety outcomes are calculated by the outcomes team based on information from the claims data.

The reliability of the data used for calculating patient safety outcomes for mortality, readmissions, and length of stay is excellent. The reliability of the data used to calculate complications, costs, and payments is also high, but lower than of the data used in calculating the 30- day mortality, readmissions and LOS outcomes, due to variation across hospitals in the number and may vary across hospitals due in the number of diagnostic and procedure codes recorded in the claim. However, due to the randomized nature of the study, we would expect there to be no difference in these outcomes between the two study arms (FLEX and STAND). References regarding the reliability of the various patient safety measures are provided below.

In addition to mortality, the following outcomes measures were collected:

- *Readmission: This calculation is based on the admission and discharge dates in the Medicare claims.*
- *Length of Stay and Prolonged Length of Stay: Calculated based on the admission and discharge dates of the index claim.*
- *Complications: Calculated using the Agency for Healthcare Research and Quality Patient Safety Indicators (see below).*
- *Costs: Calculated using inpatient, revenue center, and Part B claims. We utilize a resource costing method (more details are provided below) to estimate the costs associated with accommodations (general floor and intensive care unit), the operating room, post-discharge emergency room visits, and other services indicated by the presence of Current Procedural Terminology codes (which are translated to Relative Value Units).*

• *Payments: Calculated using the payment variables that appear in the inpatient, outpatient, and Part B claims. The total amounts paid by Medicare, the beneficiary, and the primary payer are summed. Year-based adjustments for inflation are applied to the payment figures.*

The Patient Safety Indicators were calculated by the research team using SAS programs provided by the Agency for Healthcare Research and Quality, which were run on the Medicare claims. Some of the Patient Safety Indicators that were considered "postoperative" or "perioperative" were modified for use with the study's population of medical patients. For these Patient Safety Indicators, the portion of the code that required the patient to have had surgery was deleted.

The following Patient Safety Indicators were used:

- *PSI 03 - Pressure ulcer rate*
- *PSI 06 - Iatrogenic pneumothorax rate*
- *PSI 07 - Central venous catheter-related blood stream infection rate*
- *PSI 08 - Postoperative hip fracture rate*
- *PSI 09 - Perioperative hemorrhage or hematoma rate*
- *PSI 10 - Postoperative physiologic and metabolic derangement rate*
- *PSI 11 - Postoperative respiratory failure rate*
- *PSI 12 - Perioperative pulmonary embolism or deep vein thrombosis rate*
- *PSI 13 - Postoperative sepsis rate*

References describing the validity of the claims data used to calculate the patient safety outcomes and also cited in the manuscript:

Readmissions:

- *Patel MS, Volpp KG, Small DS, et al. Association of the 2011 ACGME resident duty hour reforms with mortality and readmissions among hospitalized Medicare patients. JAMA 2014;312:2364-73.*
- *Krumholz HM, Lin Z, Keenan PS, et al. Relationship between hospital readmission and mortality rates for patients hospitalized with acute myocardial infarction, heart failure, or pneumonia. JAMA 2013;309:587-93.*

Length of Stay and Prolonged Length of Stay:

- *Silber JH, Rosenbaum PR, Even-Shoshan O, et al. Length of stay, conditional length of stay, and prolonged stay in pediatric asthma. Health Serv Res 2003;38:867-86.*
- *Silber JH, Rosenbaum PR, Kelz RR, et al. Medical and financial risks associated with surgery in the elderly obese. Ann Surg 2012;256:79-86.*
- *Silber JH, Rosenbaum PR, Koziol LF, et al. Conditional length of stay. Health Serv Res 1999;34:349-63.*
- *Silber JH, Rosenbaum PR, Rosen AK, et al. Prolonged hospital stay and the resident duty hour rules of 2003. Med Care 2009;47:1191-200.*

Patient Safety Indicators (description of PSIs):

- *Rosen AK, Loveland SA, Romano PS, et al. Effects of resident duty hour reform on surgical and procedural patient safety indicators among hospitalized Veterans Health Administration and Medicare patients. Med Care 2009;47:723-31.*

Patient Safety Indicators (validity):

- *Romano PS, Geppert JJ, Davies S, et al. A national profile of patient safety in US hospitals. Health Aff. 2003;22:154-166.*
- *Rosen AK, Rivard P, Zhao S, et al. Evaluating the patient safety indicators: how well do they perform on Veterans Health Administration data? Med Care. 2005;43:873-884.*
- *Zhan C, Miller MR. Excess length of stay, charges, and mortality attributable to medical injuries during hospitalization. JAMA. 2003;290:1868-1874.*

28. Are there any limitations you were willing to accept a priori? These could be described.

See responses to #11 and #17.

VERSION 2 – REVIEW

REVIEWER	Anne Linker, MD Mount Sinai Hospital, Division of Hospital Medicine Icahn School of Medicine USA
REVIEW RETURNED	29-May-2018

GENERAL COMMENTS	Many of my concerns from my prior review have now been addressed. The following are minor suggestions, mostly related to grammar. 1. Methods – Typo: Line 5 of paragraph, “calculated for each program as the difference, 30” – I wonder if this should read “calculated for each program as the difference IN 30 day”2. Methods – Typo: Line 6 of paragraph, should “pre intervention” be hyphenated as it is a compound adjective?3. Article Summary – Typo: Line 3 of section – should “trainees sleep” be “trainees’ sleep” (include apostrophe)4. Data Sources – Medicare fee for service is hyphenated (fee-for-service) on medicare.gov5. Time motion sub-study: line 4 – community-based should be hyphenated as it is a compound adjective6. There are multiple areas where formatting is not consistent. For instance “end-of-year surveys” is not capitalized at the start of page 19 in word document, but is capitalized in “End-of-Shift” surveys on page 18 (at the top of the page). I recommend the authors review the document to make sure formatting and grammar are consistent throughout7. In the appendix of the PDF upload, the "Figure 1 and Figure 2" (consort diagram and trial timeline) headings are not actually near the figures, as there is other information before the figures appear
---

REVIEWER	Pedro Ramos Karolinska Institutet, Sweden
REVIEW RETURNED	31-May-2018

GENERAL COMMENTS	The reviewer completed the checklist but made no further comments.
--

VERSION 2 – AUTHOR RESPONSE

Thank you for the responses. We have made all changes suggested by reviewer 3:
On page 4 we have replaced the comma with 'in' and added pre (with a hyphen) to intervention
In the article summary we have changed trainees to trainees'
In data sources we hyphenate fee-for-service (and have checked this throughout)
In time-motion section we now hyphenate community-based
We have reviewed the document for consistent capitalization and hyphen usage for end-of-shift and end-of-year
We moved the Figure captions to the end of the manuscript to they appear right before the figures